

# EDIL3 is a potential prognostic biomarker that correlates with immune infiltrates in gastric cancer

Bin Ke[1], Zheng-Kai Liang[2], Bin Li[1], Xue-Jun Wang[1], Ning Liu[1], Han Liang[1] and Ru-Peng Zhang[1]

[1] Department of Gastric Surgery, Tianjin Medical University Cancer Institute and Hospital, National Clinical Research Center for Cancer, Tianjin's Clinical Research Center for Cancer, Tianjin Key Laboratory of Cancer Prevention and Therapy, Tianjin, China

[2] Department of Gastrointestinal Surgery, Liaocheng People's Hospital, Liaocheng, Shandong, China

## ABSTRACT

**Background**. EDIL3, which contains epidermal growth factor-like repeats and discoidin I-like domains, is a secretory protein that plays an important role in embryonic development and various illnesses. However, the biological function of EDIL3 in gastric cancer (GC) is still unclear. The objective of this research was to explore the role and potential mechanism of EDIL3 in GC.

**Methods**. In this study, we used the GEPIA, HPA, MethSurv, SMART, STRING, GeneMANIA, LinkedOmics TIMER, TIMER2.0, TISIDB, and RNAactDrug databases to comprehensively analyze the roles of EDIL3 in GC. To validate the *in silico* findings, EDIL3 expression was measured in our collected GC tissues. Meanwhile, several *in vitro* experiments were performed to test the function of EDIL3 in GC.

**Results**. We found that EDIL3 was highly expressed in GC and associated with adverse clinical features. *In vitro* assays revealed that EDIL3 promoted the proliferation, migration, and invasion of GC cells. The functions of EDIL3 and co-expression genes were significantly associated with extracellular structure organization and matrix receptor interaction. EDIL3 expression was positively associated with numerous tumor-infiltrating immune cells and their biomarkers.

**Conclusion**. This study determined that EDIL3 may function as an oncogene and is associated with immune infiltration in GC. EDIL3 could be used as a potential therapeutic target for GC.

Corresponding authors
Han Liang, tj_lianghan@126.com
Ru-Peng Zhang, zhangrp65@163.com

## INTRODUCTION

Gastric cancer (GC) is the fifth most common cancer and the third leading cause of cancer-associated death worldwide, with much higher frequencies in East Asia (*Sung et al., 2021*). Although recent improvements have been achieved in the diagnosis and therapy of GC, clinical outcomes remain inadequate owing to high recurrence rates and metastatic potential (*Rawla & Barsouk, 2019*). Deciphering the underlying mechanisms of the occurrence and development of GC may result in improved diagnostic and treatment

strategies. Thus, there is an urgent need to find novel biomarkers for GC diagnosis, therapy, and prognosis.

Recently, the extracellular matrix protein have been implicated in cancer initiation and progression (*Gan et al., 2018*; *Walker, Mojares & Del Rio Hernandez, 2018*). Epidermal growth factor-like repeats and discoidin I-like domains 3 (EDIL3), also known as developmental endothelial locus-1 (DEL-1), is an extracellular matrix protein secreted by endothelial cells (*Hidai et al., 1998*). The EDIL3 protein contains two discoidin domains and three epidermal growth factor (EGF) domains, and plays many critical roles in a number of pathological and physiological processes, such as embryonic development, angiogenesis, inflammation, and immune reactions (*Klotzsche-von Ameln et al., 2017*; *Kourtzelis et al., 2019*; *Shen et al., 2017*).

Recently, the role of EDIL3 in cancer occurrence and development has attracted a great deal of attention. Expression analysis revealed that EDIL3 expression was elevated in many types of tumors, including hepatocellular carcinoma, pancreatic adenocarcinoma, colorectal cancer, and breast cancer (*Jiang et al., 2016*; *Lee et al., 2018*; *Sun et al., 2010*; *Villar-Vazquez et al., 2016*), while the expression of EDIL3 decreased in non-small cell lung cancer (*Lee et al., 2015*). Those studies indicated that EDIL3 was differentially expressed in different tumors and played different functions in different tumors.

Presently, very few studies have investigated the role of EDIL3 in GC. *Zhang et al. (2020)* reported that EDIL3 can enhance GC cell proliferation, invasion, and EMT, and the transforming growth factor-$\beta$1 (TGF-$\beta$1) signaling pathway may be involved in these processes. However, the prognostic value of EDIL3 in GC and its possible immune mechanisms are still elusive. In this research, we evaluated the expression status and clinical significance of EDIL3 in GC. Additionally, we conducted comprehensive analyses of EDIL3 in GC using multiple public databases. The results of this research may be helpful to further elucidate the function of EDIL3 in GC.

## MATERIALS & METHODS

### Tissue specimens

Fresh and paraffin-embedded GC tissue samples were obtained from Tianjin Medical University Cancer Institute and Hospital. A total of 152 primary GC specimens were obtained from patients undergoing radical gastrectomy between January 2013 and December 2015. Inclusion criteria were as follows: (1) not receiving neo-adjuvant therapy, (2) histologically proven adenocarcinoma, (3) no distant metastasis, (4) undergoing R0 resection with D2 lymph node dissection, (5) number of dissected lymph nodes greater than 15, and (6) follow-up data were complete. The surgical procedures were performed following the Japanese Gastric Cancer Association guidelines (*Japanese Gastric Cancer Association, 2017*). Clinical and histopathological data of all the cases were collected and analyzed. In addition, in order to examine the expression of EDIL3 mRNA and protein in GC, 20 pairs of GC tissues and the corresponding non-neoplastic tissues were collected between July 2020 and December 2020. All fresh tissue samples were immediately stored at −80 °C for further analysis. The matched adjacent non-neoplastic tissues were more

than five cm away from the tumor edge. This study was ratified by the Ethics Committee of Tianjin Medical University Cancer Institute and Hospital.

## qRT-PCR assay

Total RNA was extracted with Trizol reagent (Invitrogen, Carlsbad, CA, USA). RNA was reverse transcribed into complementary DNA (cDNA) using PrimeScript Kit (Takara, Shiga, Japan). The real-time quantitative PCR reactions were performed using an ABI 7900 qPCR System (Applied Biosystems, Foster City, CA, USA). Primers used for PCR were as follows: EDIL3 forward: 5′-TGACAGATGGCCGTGGATT-3′, EDIL3 reverse: 5′-TCCTCTTGGCTCCTTGGGTAA-3′; GAPDH forward: 5′-GCACCGTCAAGGCTGAGAAC-3′, GAPDH reverse: 5′-TGGTGAAGACGCCAGTGGA-3′. The conditions of PCR were as follows: 3 min at 95 °C; 40 cycles of 5 s at 95 °C 30 s at 58 °C and 1 min at 65 °C. All qRT-PCR reactions were carried out in duplicate.

## Western blot analysis

The frozen tissue samples were homogenized in RIPA lysis buffer (Beyotime Biotechnology, Shanghai, China). Protein samples were separated by 10% SDS-PAGE gel and electrotransferred to PVDF membranes. The membranes were incubated using the primary anti-EDIL3 antibody (ab151308; Abcam, Cambridge, MA). The membranes were probed with a HRP-coupled secondary antibody (Zhongshan Jinqiao Biotechnology, Beijing, China). The bands were visualized using ECL reagents. Protein expression was semi-quantified using Quantity One software (Bio-Rad, Hercules, CA, USA).

## Immunohistochemistry (IHC)

A traditional IHC staining protocol was used in this study (*Ke et al., 2020*). Tissue sections were deparaffinized and rehydrated. Antigen retrieval treatment was performed using citrate buffer, and endogenous peroxidase was blocked using 3% $H_2O_2$. The sections were incubated overnight with primary anti-EDIL3 antibody (ab151308; Abcam, Cambridge, MA), and then incubated with the corresponding secondary antibody. Staining was visualized with DAB and counterstained with hematoxylin.

The degree of immunohistochemical staining was evaluated by two independent observers according to the intensity of staining and the percentage of positive cells. The intensity of staining was scored as follows: 0 (no staining), 1 (light yellow), 2 (yellowish brown), and 3 (brown). The percentage of cells stained was scored as follows: 0 (<5%), 1 (5–25%), 2 (26–50%), 3 (51–75%), and 4 (76–100%). The final score was calculated as the multiplication of the two sores. The samples with a final staining score of ≥3 were classified as high expression and less than 3 were classified as low expression.

## Cell culture and treatment

Human GC cell lines (AGS and BGC-823) were obtained from the Chinese Academy of Science (Shanghai, China). The cells were cultured in DMEM containing 10% fetal bovine serum (FBS) at 37 °C in 5% $CO_2$. To generate EDIL3 downregulation cells, we applied lentivirus-based short hairpin RNA (shRNA) to knockdown EDIL3 expression (GeneChem, Shanghai, China). The target sequence for EDIL3 was as follows: 5′-GGAGGTTGCATCAGATGAAGA-3′ (Sh-EDIL3). The sequence for the control shRNA

was 5′-TTCTCCGAACGTGTCACGT-3′ (Sh-NC). Cells were cultured in six-well plates and treated with lentivirus for 48 h. Stable infection cells were then selected using puromycin. The effectiveness of the knockdown of EDIL3 was evaluated using qRT-PCR.

## Cell proliferation assay

Cell proliferation was assessed using Cell Counting Kit-8 (CCK-8) (Abcam, Cambridge, MA) following the manufacturer's instructions. Transfected cells were cultured in 96-well plates. Cells were incubated with 10 µl CCK-8 solution at the indicated time point after transfection, and the cell proliferation curves were constructed by measuring the 450 nm absorbance.

## Colony formation assay

GC cells were cultured into six-well plates at 500 cells per well and incubated for 10 days. The formed colonies were fixed using 4% paraformaldehyde and then stained with 0.5% crystal violet. Images of the colonies were captured and counted.

## Migration and invasion assays

The cell migration and invasion abilities were assayed using the transwell chambers method (Corning, Somerville, MA, USA). For the migration assay, cells were plated in the top chamber with serum-free DMEM, and DMEM supplemented with FBS was used as chemoattractant in the bottom chamber. After 24 h of incubation, cells on the lower surface of the top chamber were fixed and stained. The number of migrated cells were imaged and counted. For invasion assay, steps were the same as migration assay, except that the top chamber was pre-coated with Matrigel (BD, Franklin Lakes, NJ, USA).

## GEPIA database analysis

The Gene Expression Profiling Interactive Analysis (GEPIA) (http://gepia.cancer-pku.cn/) is a public database of high-throughput RNA sequencing data that can be used to analyze the RNA sequencing expression data of tumors from The Cancer Genome Atlas (TCGA) and the Genotype-Tissue Expression (GTEx) projects (*Tang et al., 2017*). In this study, we utilized the GEPIA dataset to compare the expression levels of EDIL3 in GC tissues with normal gastric tissues. In the module "Expression DIY", the correlation between EDIL3 expression and tumor stage was also investigated. Additionally, in the module "Correlation", the relationships between EDIL3 expression and multiple markers for immune cells were also investigated.

## HAP database analysis

The Human Protein Atlas (HPA) database (https://www.proteinatlas.org/) is a Swedish-based tool with the objective of mapping the location of proteins encoded by expressed genes in human tissues and cells (*Uhlen et al., 2015*). In our study, we used the HPA database to assess the EDIL3 expression at the protein level in GC tissues.

## Kaplan–Meier plotter database analysis

The Kaplan Meier plotter database (http://kmplot.com/analysis/) is a public database designed to evaluate the prognostic value of gene expression on survival in various tumor

tissues (*Lanczky & Gyorffy, 2021*). The Kaplan–Meier plotter was used to evaluate the correlation between EDIL3 expression and GC prognosis. Based on the median expression of EDIL3, GC patient samples were separated into two groups to analyze overall survival (OS) with hazard ratio (HR) with 95% confidence interval (95% CIs) and log-rank *p*-value. Additionally, we further analyzed the prognostic relevance of EDIL3 expression in the related immune cells subgroup.

## MethSurv analysis

The MethSurv (https://biit.cs.ut.ee/methsurv/) is an online tool dedicated to survival analysis based on DNA methylation data from TCGA (*Modhukur et al., 2018*). The DNA methylation data of EDIL3 in TCGA-STAD cohort and the prognostic significance of individual CpG sites were analyzed using MethSurv.

## SMART analysis

The Shiny Methylation Analysis Resource Tool (SMART) (http://www.bioinfo-zs.com/smartapp/) is an online database used to comprehensively analyze the promoter methylation status (*Li, Ge & Lu, 2019*). SMART was used to analyze the methylation levels of EDIL3 from the TCGA database. The relationship between the methylation of individual CpG sites and EDIL3 mRNA expression was also analyzed using SMART.

## LinkedOmics database analysis

The LinkedOmics database (http://www.linkedomics.org/login.php) is an online database commonly used to analyze cancer multi-omics data (*Vasaikar et al., 2018*). RNA-seq data from the TCGA-STAD dataset were selected for analysis. The LinkFinder module was used to screen for the differentially expressed genes related to EDIL3 in GC, and the correlation of results were analyzed using the Pearson correlation coefficient and visualized by heat map and volcano plot. EDIL3-associated genes were then differentially expressed and annotated using Gene Ontology (GO) and Kyoto Encyclopedia of Genes and Genomes (KEGG) pathway analysis.

## STRING database analysis

STRING (http://www.string-db.org/) is an open online database that is used to analyze and predict protein-protein interactions (PPI) (*Szklarczyk et al., 2021*). We used STRING to analyze the protein interaction and construct a PPI network, choosing the options "EDIL3" and "Human (Organism)". The minimum required interaction score was set to 0.400.

## GeneMANIA database analysis

GeneMANIA (http://genemania.org/) is an online database that helps predict the functions of favorite genes and gene-gene interaction (GGI) networks (*Mostafavi et al., 2008*). GeneMANIA was used to build an interaction network for EDIL3.

## Immune infiltration analysis by single-sample GSEA (ssGSEA)

The association between EDIL3 expression and the enrichment of immune cells in GC was analyzed using the GSVA package in R with the ssGSEA method. The relationships between EDIL3 expression and each cell immune infiltrate in GC were analyzed using Spearman

correlation analysis. In addition, the activities of 13 immune-associated functions were also evaluated based on the ssGSEA method.

## TIMER and TIMER2.0 database analysis

The Tumor IMmune Estimation Resource (TIMER) database (https://cistrome.shinyapps.io/timer/) is an online web tool used for the systematical analysis of immune infiltrates in various cancers (*Li et al., 2017*). TIMER2.0 (http://timer.cistrome.org/) is the latest version of TIMER (*Li et al., 2020a*). The "Gene" module of TIMER was used to determine the associations between EDIL3 expression and immune cell infiltration in GC. The relationships between EDIL3 expression and the multiple gene markers were analyzed using "Correlation" modules.

## TISIDB database analysis

The Tumor-Immune System Interaction Database (TISIDB) (http://cis.hku.hk/TISIDB/index.php) is an online tool used for the analysis of tumor and immune system interactions (*Ru et al., 2019*). In this study, the TISIDB database was used to evaluate the correlations between EDIL3 expression and lymphocytes, immunomodulators, and chemokines. A 'rho' value greater than 0.2 and less than −0.2 was considered a significant correlation at $P < 0.05$.

## RNAactDrug database analysis

The RNAactDrug database (http://bio-bigdata.hrbmu.edu.cn/RNAactDrug/index.jsp) is an online tool used to find correlations between drug sensitivity and RNA molecules. The correlations between drug sensitivity and EDIL3 at the expression, CNV, mutation, and methylation level were analyzed. All drugs were analyzed using three common methods, including CellMiner, GDSC, and CCLE. Pearson and Spearman correlation analysis were applied to assess the associations between EDIL3 mRNA and drug sensitivity.

## Follow-up and statistical analysis

After surgery was completed, all patients were followed up with according to the Japanese Gastric Cancer Association guidelines. Every follow-up included a physical examination, laboratory test, and radiological screening. The date of the final follow-up was December 2021.

Paired sample $t$-test was used to assess EDIL3 expression in GC tissues compared to matched adjacent non-neoplastic tissues. The Chi-square test was used to analyze relationships between EDIL3 expression and clinicopathological features. Survival analysis was applied *via* the Kaplan–Meier method and compared using log-rank test. Multivariate analysis of OS was performed using Cox regression models. Pearson's test was applied to explore the correlations between the expressions of different markers. $P$-values <0.05 were considered statistically significant. All statistical analyses were conducted using SPSS 22.0 software (IBM, USA) and R software.

## RESULTS

### Up-regulation of EDIL3 expression in GC

To investigate EDIL3 expression patterns in different tumor types, we used the online GEPIA database (Fig. 1A). The results indicated that EDIL3 expression was increased in some tumors. As shown in Fig. 1B, EDIL3 expression was elevated in GC compared with normal stomach tissues, although the differences were not statistically different. Moreover, the ''Pathological Stage Plot'' module of GEPIA was utilized to investigate the correlations between EDIL3 expression and the pathological stages of GC. Results showed a significant direct association between EDIL3 expression and GC stage (Fig. 1C, $F = 2.97$, $P = 0.0317$).

To validate these results, we applied qRT-PCR and Western blot assay to analyze the mRNA and protein levels of EDIL3 in 20 pairs of GC tumor tissues and adjacent non-neoplastic tissues. The mRNA level of EDIL3 was markedly increased in 13 (65.0%) tumor tissues compared with adjacent non-neoplastic tissues ($P = 0.0401$, Fig. 1D). Western blotting results showed an EDIL3 band at the expected size of 54 kDa. Consistent with qRT-PCR data, EDIL3 protein expression was also increased in 12 (60.0%) GC tissues ($P = 0.0119$, Figs. 1E and 1F). Of these 12 patients, the mRNA and protein expression levels were both increased in tumor tissues. The results of our study were concordant with those in the GEPIA database.

### Correlations between EDIL3 expression and clinicopathological characteristics in GC

To investigate the clinical role of EDIL3 in GC, we examined its expression in GC tissues using immunohistochemistry. Our data revealed that EDIL3 was mainly expressed in the cytoplasm of GC cells (Fig. 1G). Among the 152 tested GC tissues, 57.9% (88/152) were positive for EDIL3 expression, and 64 (42.1%) cases were negative. As shown in Table 1, positive expression of EDIL3 was associated with deeper tumor depth ($P = 0.041$), lymph node metastasis ($P = 0.002$), and advanced pTNM stage ($P = 0.023$). EDIL3 expression did not correlate with gender, age, tumor size, or tumor differentiation ($P > 0.05$).

In order to evaluate the prognostic significance of EDIL3 in GC, the Kaplan–Meier method was used to analyze the relationship between EDIL3 expression and OS. The results showed that the high expression of EDIL3 was negatively associated with OS ($P < 0.001$, log-rank test, Fig. 2A). To validate our findings, the Kaplan–Meier plotter and HPA databases were used to assess the prognostic role of EDIL3 in GC. As shown in Figs. 2B and 2C, similar results were obtained.

To further investigate the role of EDIL3 in clinical prognosis, univariate and multivariate analyses of clinical follow-up data were carried out. Univariate Cox analysis showed that serosal invasion, lymph node metastasis, pTNM stage, and EDIL3 expression were notably correlated with OS in GC (Fig. 2D). Multivariate Cox analysis demonstrated that lymph node metastasis, pTNM stage, and EDIL3 expression were independent factors of OS in our cohort (Fig. 2E). Overall, these results demonstrated that EDIL3 was closely associated with tumor progression and may serve as a potential prognostic marker for GC.

Peer J

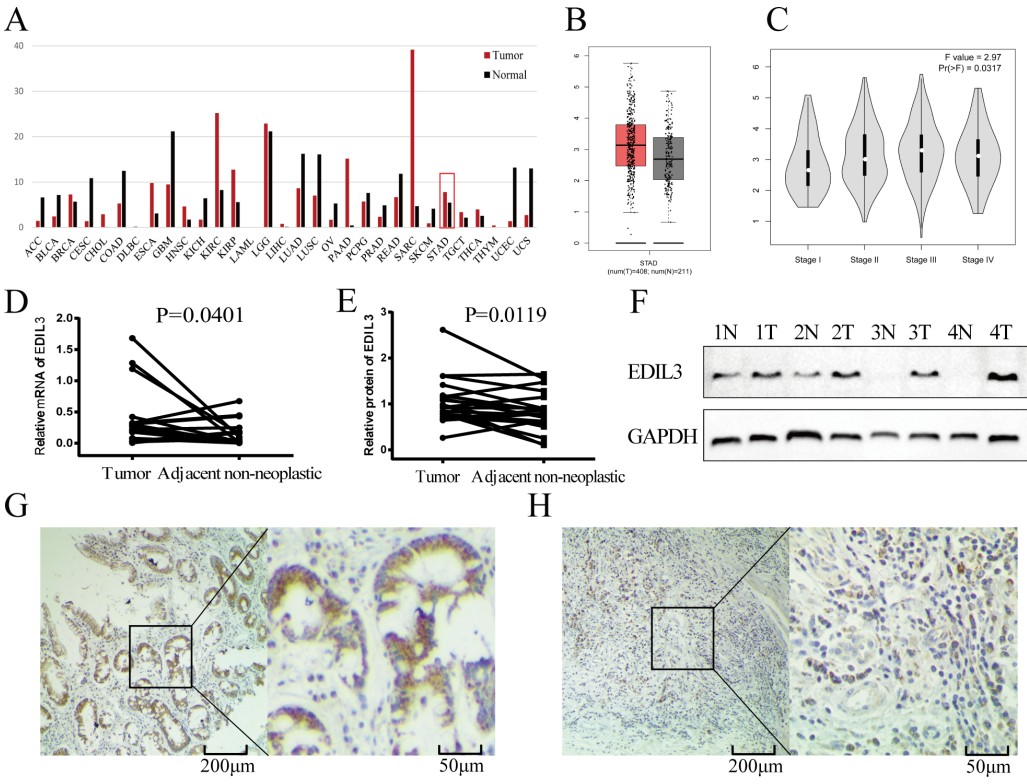

**Figure 1 EDIL3 is up-regulated in GC.** (A) EDIL3 expression in different tumor types were analyzed by GEPIA. (B) EDIL3 expression level in tumor tissues and normal stomach tissues determined by GEPIA. (C) The relationship between EDIL3 expression and the pathological stages of GC using GEPIA. (D) Relative EDIL3 mRNA expression levels in tumor tissues and adjacent non-neoplastic tissues. (E) Relative EDIL3 protein expression levels in tumor tissues and adjacent non-neoplastic tissues (EDIL3/GAPDH). (F) Representative result of EDIL3 protein expression in four paired GC tissues and adjacent non-neoplastic tissues. (G) Representative IHC image of EDIL3 high expression in GC. (H) Representative IHC image of EDIL3 low expression in GC. T: GC tumor tissues, N: Adjacent non-neoplastic tissues.

## The malignant behaviors of EDIL3 in GC

In order to further evaluate the potential role for EDIL3 in the tumorigenesis and development of GC, we performed several cellular function assays. After lentiviral transduction, EDIL3 was knocked down successfully in AGS and BGC-823 cells (Fig. 3A). The results of CCK-8 assays indicated that the proliferation ability of AGS and BGC-823 cells was strongly restrained by EDIL3 knockdown (Fig. 3B). Consistently, clone formation experiments also indicated that EDIL3 knockdown can inhibit the cloning ability of AGS and BGC-823 cells (Fig. 3C). Moreover, the results of transwell migration and invasion indicated that EDIL3 knockdown markedly inhibited the migratory and invasive capability of AGS and BGC-823 cells (*Yuan et al., 2019*; see their Figure 2). Collectively, these findings demonstrated that EDIL3 functioned like an oncogene and promoted malignant behaviors *in vitro* in GC.

**Table 1  The relationship between the expression of EDIL3 and clinicopathological parameters in GC.**

| Clinicopathological parameter | Total | Expression of EDIL3 | | $\chi^2$ | $p$ |
|---|---|---|---|---|---|
| | | Low | High | | |
| Gender | | | | | |
| Male | 106 | 48 | 58 | 1.541 | 0.228 |
| Female | 46 | 16 | 30 | | |
| Age | | | | | |
| ≤60 | 77 | 35 | 42 | 0.718 | 0.397 |
| >60 | 75 | 29 | 46 | | |
| Tumor size | | | | | |
| ≤5cm | 85 | 32 | 53 | 1.572 | 0.210 |
| >5cm | 67 | 32 | 35 | | |
| Tumor differentiation | | | | | |
| Well/moderate | 44 | 17 | 27 | 0.306 | 0.580 |
| Poor/undifferentiation | 108 | 47 | 61 | | |
| Serosal invasion | | | | | |
| Negative | 48 | 26 | 22 | 4.187 | 0.041 |
| Positive | 104 | 38 | 66 | | |
| Lymph node metastasis | | | | 9.183 | 0.002 |
| No | 36 | 23 | 13 | | |
| Yes | 116 | 41 | 75 | | |
| pTNM stage | | | | | |
| I–II stage | 51 | 28 | 23 | 5.156 | 0.023 |
| III stage | 101 | 36 | 65 | | |

## DNA methylation of EDIL3 in GC

Abnormal DNA methylation can influence gene expression and play important roles in tumor initiation and progression. The results of SMART analysis showed that the methylation status of EDIL3 genes was different in different cancers, suggesting that it may play distinct roles in different tumors (Fig. 4A). There was no apparent difference between the methylation status of EDIL3 in GC tissues and normal tissues.

To explore the mechanism of methylation, we utilized the MethSurv database to assess the relationship between EDIL3 and all CpG sites. As shown in Fig. 4B, there were 20 predicted CpG sites of EDIL3. Among these CpG sites, 18 CpG sites had decreased methylation (Table 2), and cg17978562 showed the lowest levels of methylation. Consistently, the low levels of DNA methylation in five CpG sites (cg05179846 (Fig. 4C), cg03478689 (Fig. 4E), cg16099804 (Fig. 4G), cg16773899 (Fig. 4I), and cg01072952 (Fig. 4K)) were meaningfully correlated with better prognosis in GC, with cg16099804 being the most pronounced ($P = 0.022$). Additionally, relationship analysis revealed that the DNA methylation levels in all five CpG sites were significantly inversely correlated with EDIL3 expression (Figs. 4D, 4F, 4H, 4J and 4L). In summary, our results demonstrated that the methylation status of EDIL3 might influence the prognosis of GC patients.

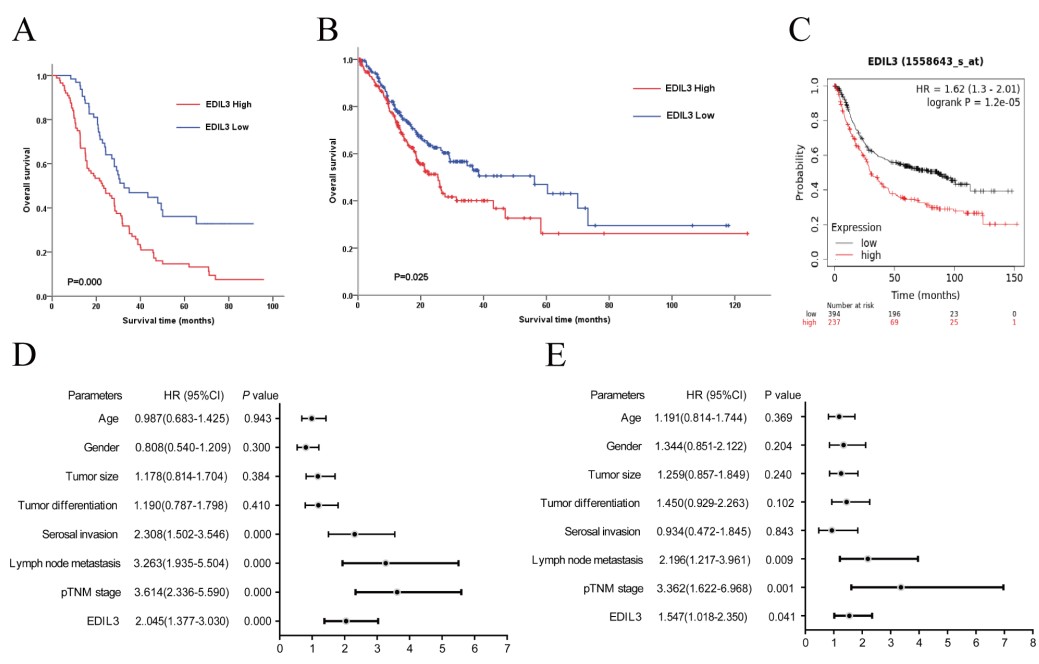

**Figure 2** **The prognostic significance of EDIL3 in GC.** (A) Survival analysis of EDIL3 expression in 152 GC patients after curative resection. (B) Survival analysis of the EDIL3 expression in GC patients by HPA database. (C) Survival analysis of the EDIL3 expression in GC patients by Kaplan–Meier plotter database. (D) Univariate analysis of parameters correlated with OS of GC patients. (E) Multivariate analysis of parameters correlated with OS of GC patients.

## Interaction network of EDIL3 in GC

To investigate the potential function of EDIL3, we constructed the PPI network for EDIL3 using the STRING database (Fig. 5A). The results showed that 20 proteins were closely related to EDIL3, including ITGAV, PTK2, ITGB3, ITGB5, and CLAC4. In addition, the GeneMANIA database was utilized to construct a GGI network (Fig. 5B). There were 20 altered genes were closely correlated with EDIL3. The top five genes most associated with EDIL3 were ITGB5, ITGAV, PTK2, PAFAH1B1, and ITGB3. Functional analysis indicated that these genes were significantly correlated with receptor complex, cell-substrate adhesion, integrin complex, and protein complex involved in cell adhesion. Moreover, the results indicated that the EDIL3 interaction network included five common EDIL3-interacting genes across the STRING and GeneMANIA databases: ITGAV, ITGB3, ITGB5, ZNF469, and PTK2. The GEPIA database showed that EDIL3 expression was directly associated with the expression of these five genes in GC (Fig. 5C).

## EDIL3 co-expression network in GC

In order to investigate the function of EDIL3 in GC, the LinkedOmics database was employed to investigate the co-expression pattern of EDIL3. The results revealed that there were 7,673 that were genes positively associated and 4,250 genes that were negatively associated with EDIL3 (Fig. 6A). Based on the Spearman test, the top 50 genes that were significantly positively and negatively associated with EDIL3 were presented using heat

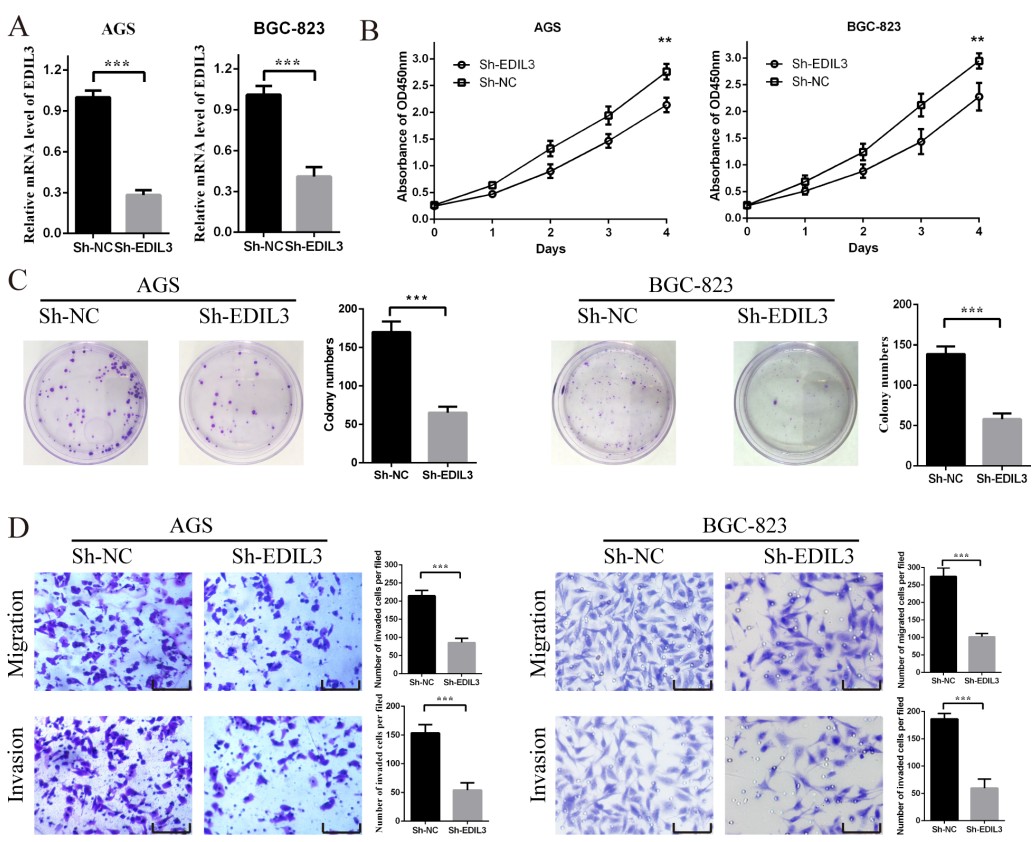

**Figure 3** **The malignant behaviors of EDIL3 in GC.** (A) EDIL3 mRNA expression in AGS and BGC-823 cells after sh-RNA transfection. (B) The cell viability of AGS and BGC-823 cells measured by CCK-8 after sh-RNA transfection. (C) Colony formation assay of AGS and BGC-823 cells after sh-RNA transfection. (D)The migration and invasion assays of AGS and BGC-823 cells after sh-RNA transfection. Scale bars, 100 μm.

maps (Figs. 6B and 6C). It is worth noting that 17 of the 50 positively associated genes and three of the 50 negatively associated genes indicated markedly high and low hazard ratios (HR) ($P < 0.05$) for GC patient survival (Fig. 6D).

GO analysis revealed that EDIL3 co-expressed genes mainly participated in extracellular structure organization, endothelium development, substrate-dependent cell migration, phospholipase C-activating G protein-coupled receptor signaling pathway, and vasculogenesis (Fig. 6E). KEGG pathway analysis indicated that EDIL3 co-expressed genes were mainly enriched in ECM-receptor interaction, arrhythmogenic right ventricular cardiomyopathy, complement and coagulation cascades, dilated cardiomyopathy (DCM), and focal adhesion (Fig. 6F). These results demonstrated the widespread influence of EDIL3 on the occurrence and development in GC.

## Correlation analysis between EDIL3 expression and immune infiltration in GC

Many researchers have found that tumor immune infiltration is associated with tumor aggressiveness and progression in various malignancies. Here, we investigated whether

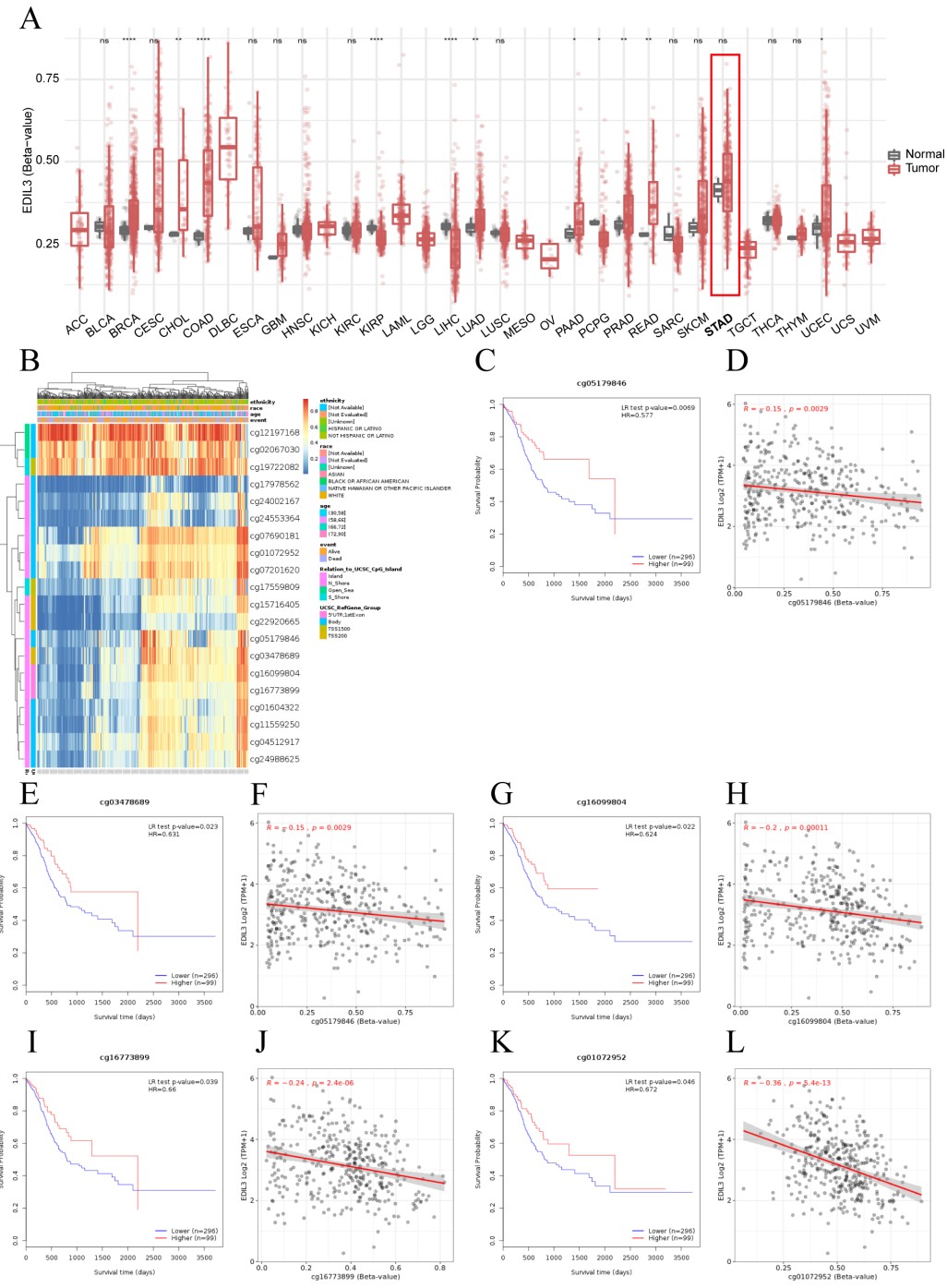

**Figure 4** **The methylation of EDIL3 in GC.** (A) The methylation levels of EDIL3 in multiple tumor types by SMART. (B) DNA methylation heat map of EDIL3 in GC by MethSurv. Red to blue: high levels of DNA methylation to low levels. The survival analysis of methylation site cg05179846 (C), cg03478689 (E), cg16099804 (G), cg16773899 (I), and cg01072952 (K) in GC. The correlation analysis between EDIL3 mRNA expression and cg05179846 (D), cg03478689 (F), cg16099804 (H), cg16773899 (J), and cg01072952 (L) methylation in GC.

**Table 2  The prognostic values of CpG sites in EDIL3.**

| CpG Name | HR | CI | LR test p value | RefGene Group | Relation to CpG Island |
|---|---|---|---|---|---|
| cg03478689 | 0.631 | (0.415;0.957) | 0.023 | TSS200 | Island |
| cg05179846 | 0.577 | (0.378;0.88) | 0.0069 | Body | Island |
| cg07201620 | 0.727 | (0.527;1.002) | 0.051 | Body | Island |
| cg07690181 | 0.813 | (0.588;1.122) | 0.21 | Body | Island |
| cg15716405 | 0.783 | (0.523;1.17) | 0.22 | TSS200 | Island |
| cg16099804 | 0.624 | (0.409;0.954) | 0.022 | 5′UTR; 1stExon | Island |
| cg16773899 | 0.66 | (0.437;0.997) | 0.039 | 5′UTR; 1stExon | Island |
| cg22920665 | 0.813 | (0.55;1.204) | 0.29 | TSS200 | Island |
| cg24988625 | 0.736 | (0.493;1.099) | 0.12 | Body | Island |
| cg01072952 | 0.672 | (0.448;1.008) | 0.046 | Body | N_Shore |
| cg01604322 | 0.798 | (0.541;1.175) | 0.24 | Body | N_Shore |
| cg04512917 | 0.793 | (0.531;1.183) | 0.25 | Body | N_Shore |
| cg11559250 | 0.684 | (0.453;1.033) | 0.061 | Body | N_Shore |
| cg17978562 | 0.902 | (0.633;1.286) | 0.57 | Body | N_Shore |
| cg24002167 | 0.745 | (0.499;1.112) | 0.14 | Body | N_Shore |
| cg24553364 | 0.793 | (0.574;1.094) | 0.16 | Body | N_Shore |
| cg02067030 | 1.107 | (0.758;1.616) | 0.6 | Body | Open_Sea |
| cg12197168 | 1.225 | (0.855;1.753) | 0.28 | Body | Open_Sea |
| cg17559809 | 0.695 | (0.462;1.043) | 0.069 | TSS1500 | S_Shore |
| cg19722082 | 0.896 | (0.613;1.308) | 0.56 | TSS1500 | S_Shore |

EDIL3 expression was associated with immune infiltration in GC. We first evaluated the relationship of EDIL3 expression with immune cell enrichment through the ssGSEA algorithm in GC. Spearman correlation analysis indicated that EDIL3 expression was positively associated with 16 immune cell enrichments (Fig. 7A). The top five associated immune cells were mast cells ($r = 0.460$, $P < 0.001$), TFH ($r = 0.334$, $P < 0.001$), eosinophils ($r = 0.330$, $P < 0.001$), NK cells ($r = 0.325$, $P < 0.001$), and iDC ($r = 0.323$, $P < 0.001$). Additionally, we utilized TIMER to examine the correlation between EDIL3 expression and the infiltration of six major types of immune cells and tumor purity. Our assays indicated that EDIL3 expression was significantly positively associated with all six immune cells, including macrophages ($r = 0.43$, $P = 4.75e{-}18$), CD4+ T cells ($r = 0.372$, $P = 1.86e{-}13$), dendritic cells ($r = 0.271$, $P = 1.09e{-}07$), neutrophils ($r = 0.211$, $P = 4.33e{-}05$), CD8+ T cells ($r = 0.188$, $P = 2.71e{-}04$), and B cells ($r = 0.169$, $P = 1.08e{-}03$). In contrast, EDIL3 expression was inversely associated with tumor purity ($r = -0.129$, $P = 1.17e{-}02$) (Fig. 7B). As for the related immune functions, the scores for CCR, type II IFN response, major histocompatibility complex (MHC) class I, parainflammation, APC co-inhibition, and type I IFN response were significantly higher in the high EDIL3 expression group than in the low expression group ($P < 0.05$) (Fig. 7C).

Recent studies indicated that many tumors express immune checkpoints to escape immune cell attacks. Immune checkpoint proteins play critical roles in immune response regulation. We next investigated the relationship between EDIL3 and immune checkpoint

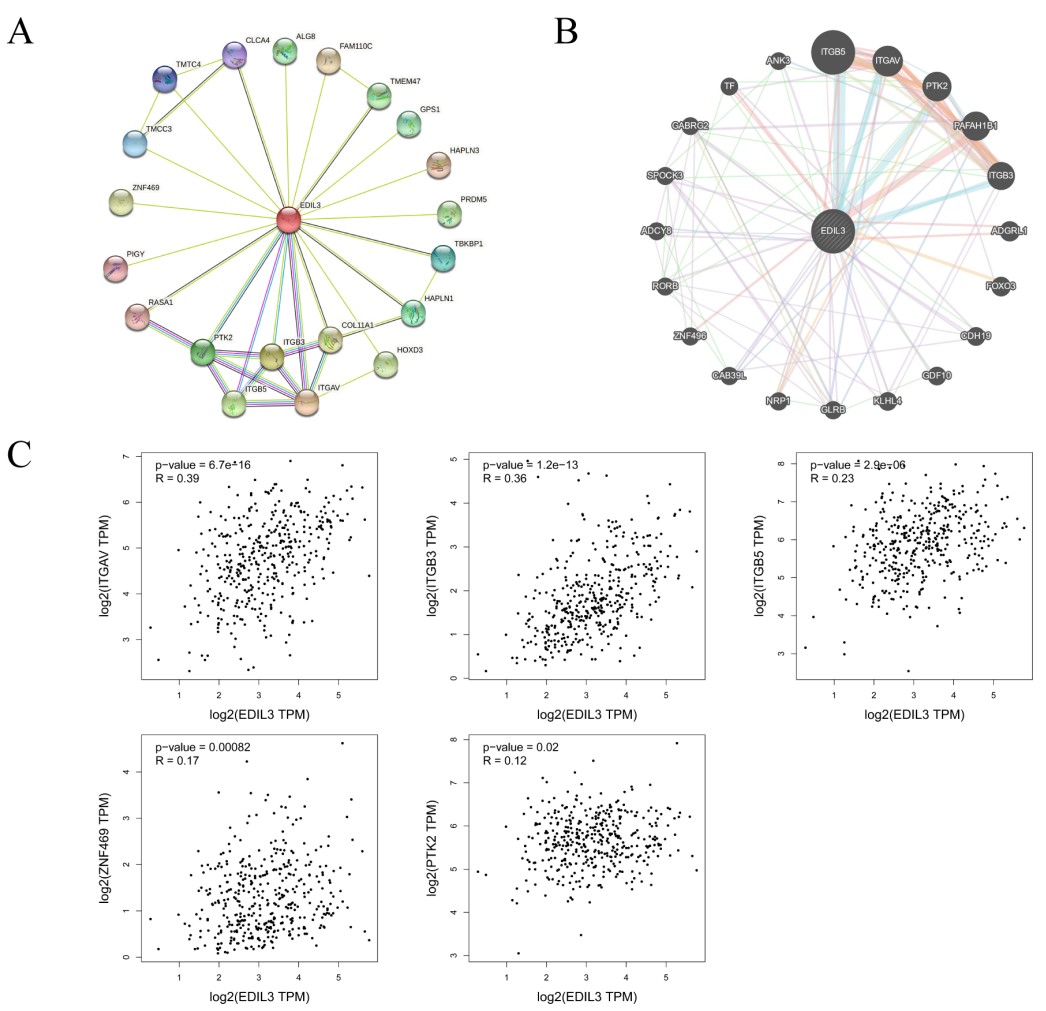

**Figure 5** **Interaction analysis of EDIL3 at the protein and gene levels.** (A) PPI network of EDIL3 constructed *via* the STRING database. (B) GGI network of EDIL3 constructed *via* GeneMANIA. (C) Scatter plots of associations between EDIL3 expression and ITGAV, ITGB3, ITBG5, ZNF469, and PTK2 in GC.

molecules using the TIMER2.0 database. We found that EDIL3 expression was positively correlated with PD-1, PD-L1, PD-L2, CTLA4, TIM-3, and TIGIT (all $P < 0.05$) and negatively correlated with IDO1 ($P = 0.0764$) and LAG-3 ($P = 0.208$) (Fig. 8A). Moreover, the co-expression analyses of EDIL3 and mismatch repair (MMR) signatures in GC were also performed. The analysis showed that EDIL3 expression was significantly positively associated with MSH6 and MLH1 (Fig. 8B). These findings demonstrate that EDIL3 plays critical roles in immune cell infiltration in GC and might affect the efficacy of immunotherapy.

## Correlation between EDIL3 expression and immune molecules

After confirming the relationship between EDIL3 expression and immune infiltration, we further validated the relationship between EDIL3 expression and the related biomarkers of different immune cells in GC using TIMER. The results indicated that EDIL3 expression had

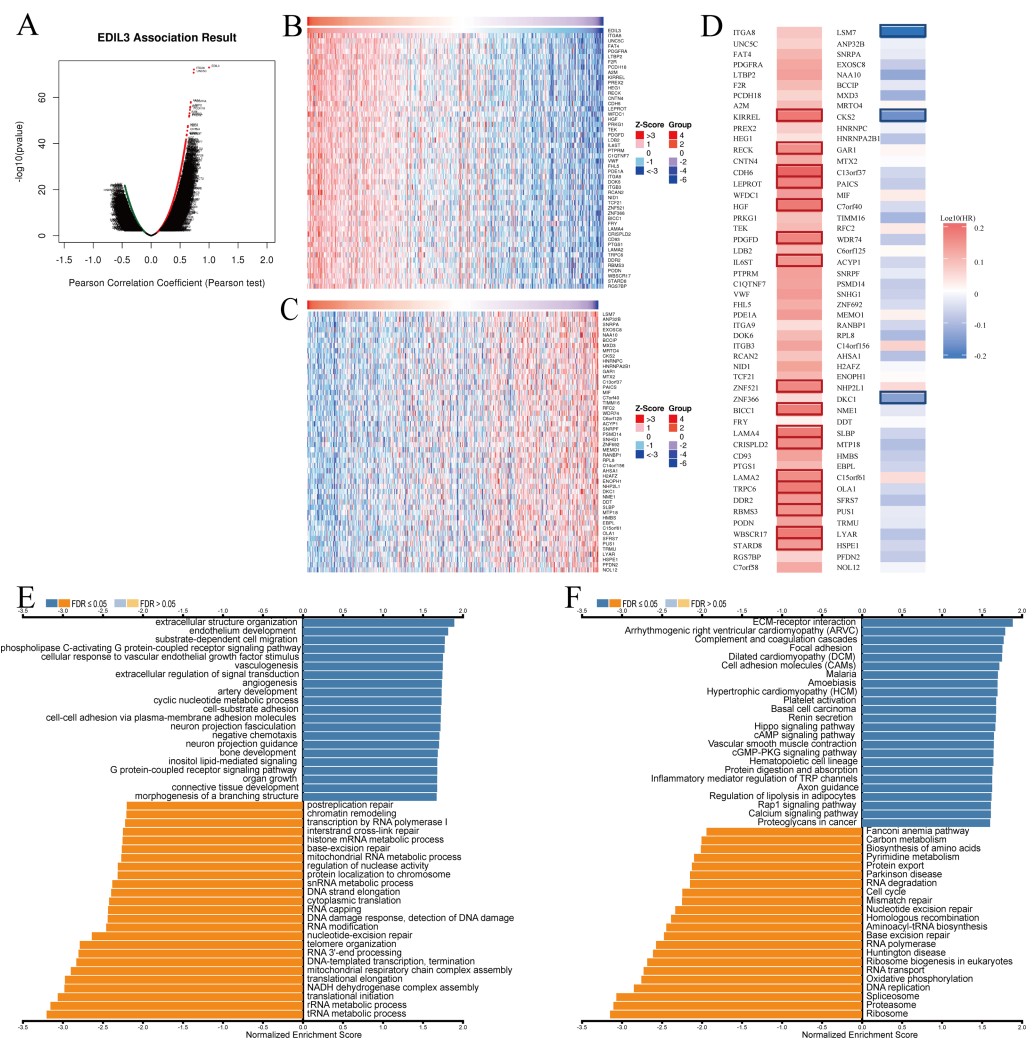

**Figure 6 The EDIL3 co-expression genes in GC.** (A) Volcano map of co-expressed genes of EDIL3 in GC cohort. (B) Heat map of the top 50 genes positively associated with EDIL3. (C) Heat map of the top 50 genes negatively associated with EDIL3. Red represents positively linked genes and blue represents negatively linked genes. (D) Survival map of the top 50 genes positively and negatively associated with EDIL3 in GC using GEPIA. The red and blue squares indicate higher and lower risks for survival, respectively. The bordered squares indicate the significant unfavorable and favorable survival ($P < 0.05$). (E) EDIL3 co-expression genes were categorized for GO analysis in GC cohort. (F) EDIL3 co-expression genes were categorized for KEGG pathway analysis in GC cohort.

significant positive correlations with most immune gene markers and these correlations barely changed after tumor purity correction (Table 3). BDCA-4 ($r = 0.492$), STAT5B ($r = 0.478$), CD1c ($r = 0.450$), CCR7 ($r = 0.403$), and TGF $\beta$1 ($r = 0.385$) were the top five correlated gene markers.

In addition, we also evaluated the correlation between EDIL3 expression and various immune components in GC from TISIDB. EDIL3 was positively associated with most tumor-infiltrating lymphocytes (TILs), including mast cells, eosinophils, macrophages, memory B cells, NK cells, and Th2 cells (Fig. 9A). Next, we explored the correlations

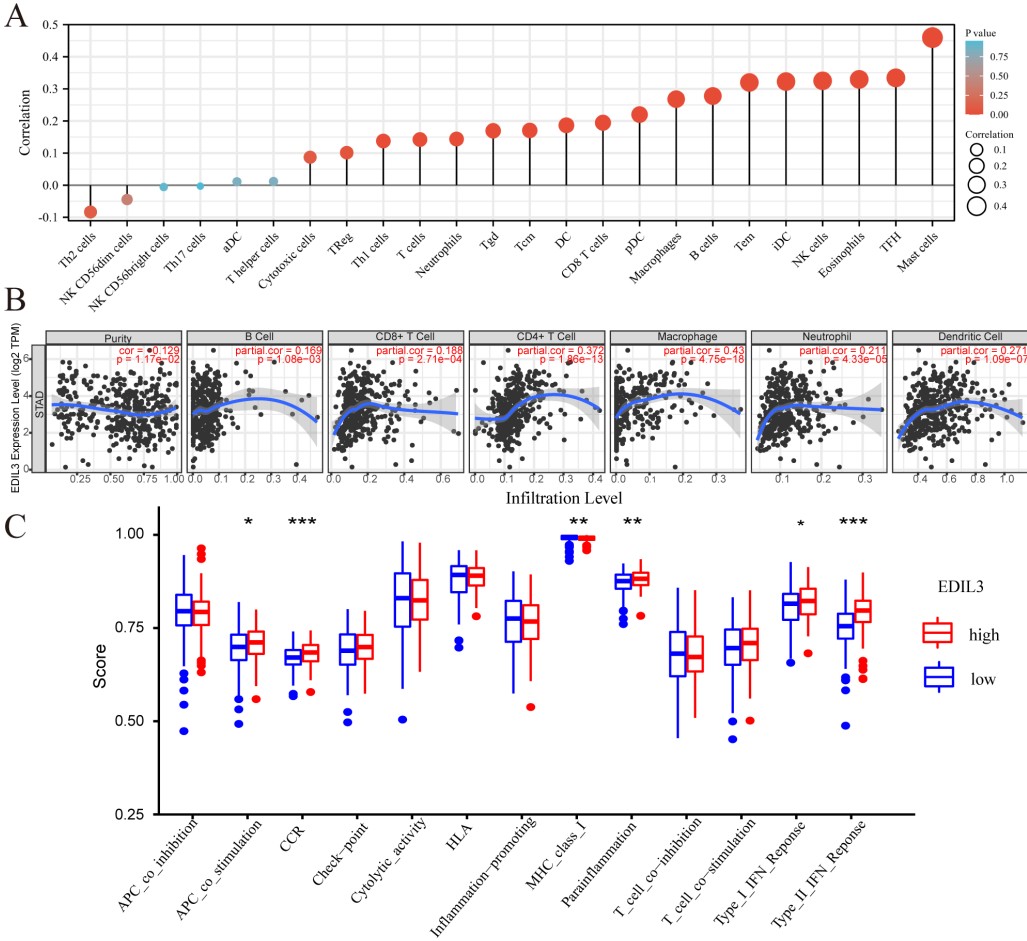

**Figure 7** **The relationship of EDIL3 expression with immune infiltration in GC.** (A) The relationship between EDIL3 expression and immune cell enrichment in GC through ssGSEA algorithm. The size of the dots represents the absolute Spearman's correlation coefficient values. (B) The relationship between EDIL3 expression and immune infiltration in GC from TIMER. (C) The relationship between risk score and different immune features in TCGA cohort. (* $P < 0.05$, ** $P < 0.01$, *** $P < 0.001$).

between EDIL3 and three different types of immunomodulators. EDIL3 expression was significantly associated with immunostimulators, such as ENTPD1_exp, TNFSF18_exp, CD28_exp, C10orf54_exp, CXCL12_exp, and TNFSF15_exp (Fig. 9B). EDIL3 expression was significantly associated with immunoinhibitors, such as KDR_exp, CSF1R_exp, TGFB1_exp, ADORA2A_exp, TGFBR1_exp, and PDCD1LG2_exp (Fig. 9C). EDIL3 expression was significantly associated with MHC molecules, such as HLA-DOA_exp, HLA-DOB_exp, HLA-DQA2_exp, HLA-DQA1_exp, and HLA-DPB1 (Fig. 9D). Finally, we investigated the associations between EDIL3 expression and chemokines and receptors. Figure 9E depicts associations between EDIL3 expression and chemokines, including CXCL14_exp, CCL11_exp, CCL14_exp, CXCL12_exp, CCL19_exp, and CCL22. Correlations between EDIL3 expression and receptors, including CX3CR1_exp, CCR4_exp, CXR1_exp, CCR7_exp, CXCR4_exp, and CCR2_exp, are shown in Fig. 9F.

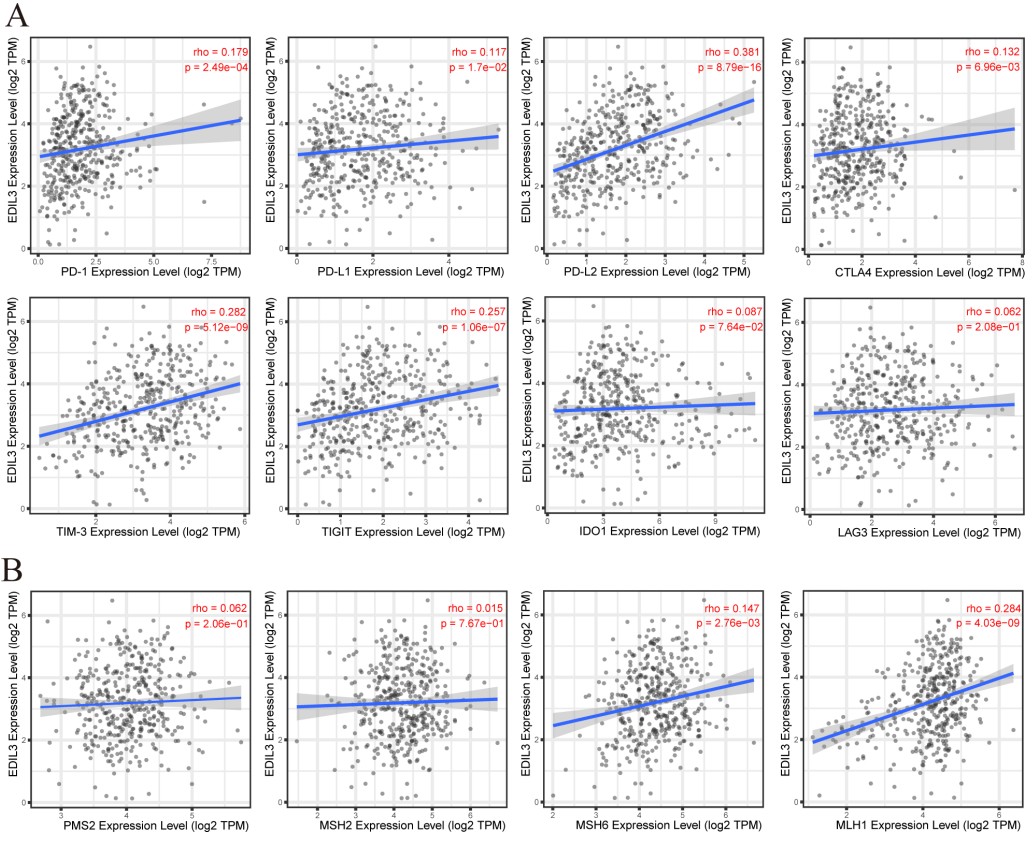

**Figure 8** **The correlation between EDIL3 and immune checkpoints, MMR in GC from TIMER 2.0.** (A) The correlation between EDIL3 expression and eight known immune checkpoint molecules. (B) The correlation between EDIL3 expression and four known MMR-related genes.

In summary, the above findings indicate that EDIL3 was widely involved in regulating multiple immune molecules in GC, thereby affecting immune infiltration.

## Prognostic significance of EDIL3 expression based on immune cells

Our results indicate that EDIL3 expression was associated with immune infiltration and various immune cell markers. Elevated EDIL3 expression was correlated with worse prognosis in GC. Therefore, we conjecture that EDIL3 affected the prognosis partly due to immune infiltration. To corroborate this hypothesis, we evaluated the prognostic value of EDIL3 in the subgroups with enriched and decreased immune cells *via* the Kaplan Meier plotter (Fig. 10). The results indicated that high EDIL3 expression was markedly correlated with worse prognosis in the enriched subgroups of CD8+ T cells, natural killer T cells, regulatory T cells, and type 2 T helper cells (Figs. 10B–10E), while there was no significant association among decreased subgroups. In contrast, high EDIL3 expression in decreased B cells and macrophages (Figs. 10F–10G) subgroups correlated with worse prognosis. There was no clear association between EDIL3 expression and prognosis in the group with different levels of CD4+ T cells and type 1 T helper cells. These data demonstrated that EDIL3 may affect prognosis partly because of immune infiltration.
**Table 3** The correlations between EDIL3 and gene markers of immune cells via TIMER.

| Description | Gene marker | None | | Purity | |
|---|---|---|---|---|---|
| | | Cor | *p* | Cor | *P* |
| B cell | CD19 | 0.296 | *** | 0.303 | *** |
| | CD79A | 0.337 | *** | 0.336 | *** |
| CD8+ T cell | CD8A | 0.212 | *** | 0.212 | *** |
| | CD8B | 0.110 | * | 0.120 | * |
| T cell (general) | CD2 | 0.242 | *** | 0.252 | *** |
| | CD3D | 0.184 | *** | 0.186 | *** |
| | CD3E | 0.218 | *** | 0.233 | *** |
| | T-bet(TBX21) | 0.190 | *** | 0.208 | *** |
| Th1 | STAT1 | 0.091 | 0.064 | 0.087 | 0.092 |
| | STAT4 | 0.301 | *** | 0.316 | *** |
| | INF-$\alpha$(TNF) | 0.143 | ** | 0.134 | ** |
| | INF-$\gamma$(IFNG) | −0.033 | 0.501 | −0.024 | 0.645 |
| Th2 | STAT5A | 0.341 | *** | 0.347 | *** |
| | STAT6 | 0.274 | *** | 0.268 | *** |
| | GATA3 | 0.207 | *** | 0.227 | *** |
| | IL 13 | 0.060 | 0.226 | 0.063 | 0.219 |
| Treg | CCR8 | 0.382 | *** | 0.382 | *** |
| | FOXP3 | 0.301 | *** | 0.305 | *** |
| | STAT5B | 0.498 | *** | 0.478 | *** |
| | TGF$\beta$1 | 0.393 | *** | 0.385 | *** |
| Tumor-associated Macrophage | CCL2 | 0.287 | *** | 0.275 | *** |
| | CD68 | 0.303 | *** | 0.290 | *** |
| | IL 10 | 0.346 | *** | 0.343 | *** |
| M1 macrophage | IRF5 | 0.241 | *** | 0.241 | *** |
| | COX2(PTGS2) | 0.232 | *** | 0.215 | *** |
| | INOS(NOS2) | 0.100 | * | 0.119 | * |
| M2 macrophage | CD163 | 0.367 | *** | 0.360 | *** |
| | VSIG4 | 0.288 | *** | 0.293 | *** |
| | MS4A4A | 0.348 | *** | 0.349 | *** |
| Neutrophil | CD11b(ITGAM) | 0.380 | *** | 0.384 | *** |
| | CD66b(CEACAM8) | 0.087 | 0.077 | 0.116 | * |
| Dendritic cell | CCR7 | 0.390 | *** | 0.403 | *** |
| | CD1c(BDCA-1) | 0.441 | *** | 0.450 | *** |
| | CD11c (ITGAX) | 0.352 | *** | 0.344 | *** |
| | BDCA-4(NRP1) | 0.508 | *** | 0.492 | *** |
| | HLA-DPB1 | 0.203 | *** | 0.205 | *** |
| | HLA-DQB1 | 0.099 | * | 0.093 | 0.070 |
| | HLA-DRA | 0.141 | ** | 0.145 | ** |
| | HLA-DPA1 | 0.171 | *** | 0.175 | *** |

| Description | Gene marker | None | | Purity | |
|---|---|---|---|---|---|
| | | Cor | *p* | Cor | *P* |
| Natural killer cell | KIR3DL3 | 0.024 | 0.624 | 0.039 | 0.452 |
| | NCR1 | 0.203 | *** | 0.222 | *** |

**Notes.**
*$P < 0.05$.
**$P < 0.01$.
***$P < 0.001$.

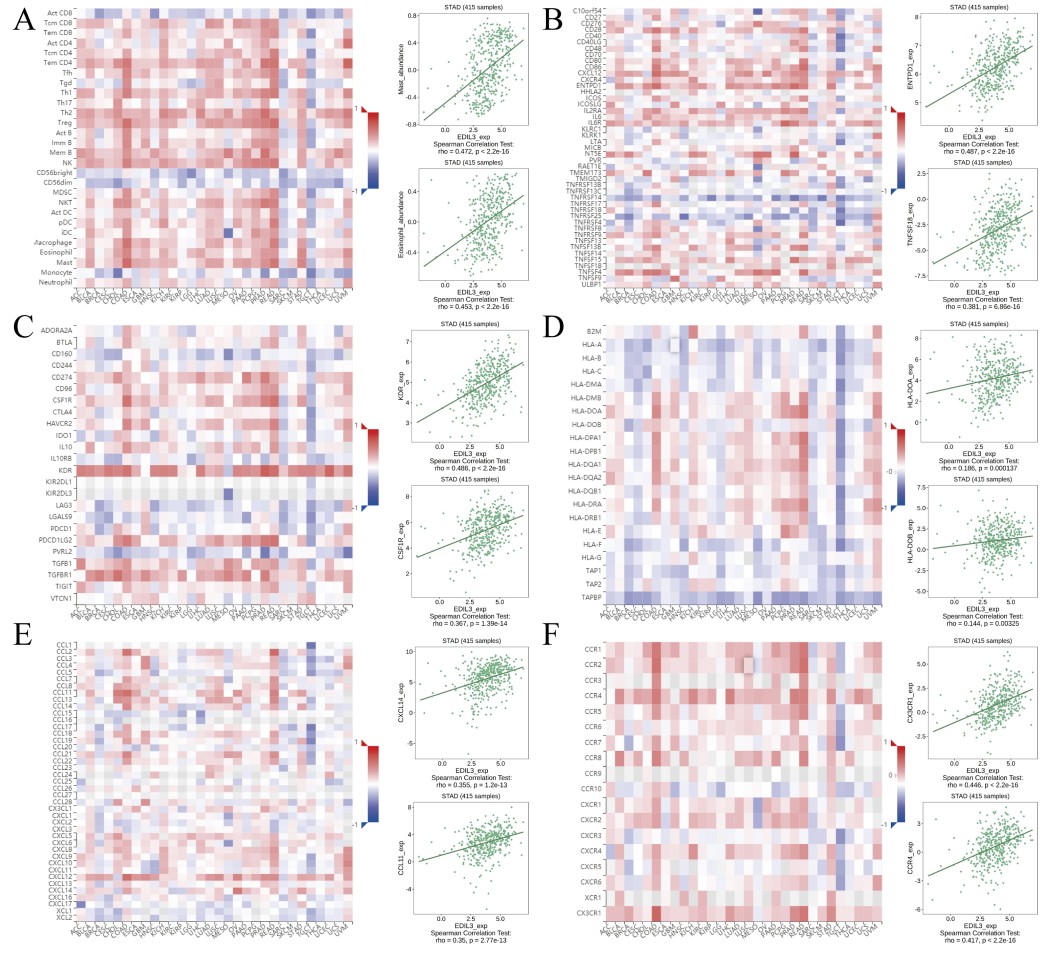

**Figure 9  Relationship between EDIL3 expression and various immune components in GC from TISIDB.** (A) Correlation between EDIL3 expression and abundance of TILs. Correlations between the EDIL3 expression and immunomodulators, including immunostimulators (B), immunoinhibitors (C), and MHC molecules (D). Correlations between EDIL3 expression and chemokines (E) and receptors (F).

## Relationship between EDIL3 expression and drug sensitivity

Among 361 drugs, we found the EDIL3 mRNA expression was strongly positively correlated with four drugs ($R > 0.3$ and fdr<0.01): n-acetylsulfanilyl chloride, melisimplexin,(E)-5-chloro-3-((6-(4-chlorophenyl)-2-cyclopropylimidazole)),  and panobinostat (Table S1). In addition, we investigated the relationship between EDIL3 and sensitivity of drugs

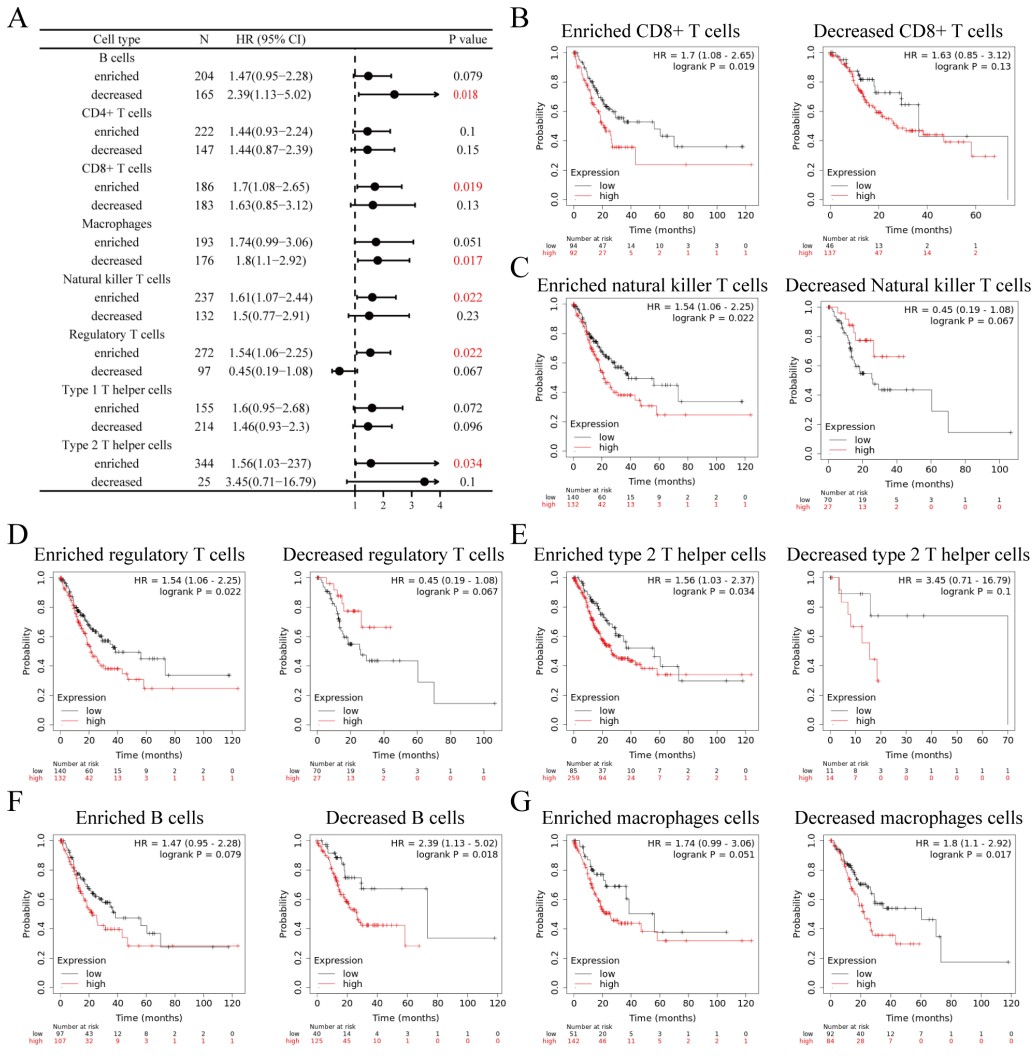

**Figure 10** **Prognostic significance of EDIL3 expression in immune cell subgroups in GC.** (A) The association between EDIL3 and OS in different immune cell subgroups in GC. Relationships between EDIL3 expression and OS in GC base on enriched or decreased CD8+ T cell (B), natural killer T cell (C), regulatory T cell (D), type 2 T helper cell (E), B cell (F), and macrophage (G) subgroups.

commonly used to treat GC. As shown in Table 4, 5-fluorouracil, paclitaxel, sorafenib, and lapatinib were positively correlated with EDIL3 mRNA expression; mitomycin-C was positively correlated with EDIL3 CNV; and docetaxel was positively correlated with EDIL3 methylation level. These results suggest that EDIL3 expression may regulate the drug sensitivity of GC and further research is needed.

## DISCUSSION

GC is one of the most common and lethal cancers in China (*Cao et al., 2021*). Comprehensive treatment based on surgery is the standard therapy modality used for GC. In recent years, systemic treatment including chemotherapy, targeted therapy, and

**Table 4  The relationship between EDIL3 mRNA and anticancer drug sensitivity by RNAactDrug.**

| Compound | RNAmolecule | Omics | Source | r-Pearson | FDR-Pearson | r-Spearman | FDR-Spearman |
|---|---|---|---|---|---|---|---|
| 5-Fluorouracil | EDIL3 | Expression | GDSC | 0.125 | *** | 0.091 | ** |
| Paclitaxel | EDIL3 | Expression | CCLE | NA | NA | 0.171 | * |
| Sorafenib | EDIL3 | Expression | CCLE | NA | NA | 0.224 | ** |
| Lapatinib | EDIL3 | Expression | CCLE | NA | NA | 0.191 | * |
| Mitomycin-C | EDIL3 | CNV | GDSC | NA | NA | 0.083 | * |
| Docetaxel | EDIL3 | Methylation | GDSC | 0.106 | ** | 0.098 | * |

**Notes.**

CNV, copy number variation; GDSC, Genomics of Drug Sensitivity in Cancer; CCLE, Cancer Cell Line Encyclopedia; NA, not available.

*$p < 0.05$

**$P < 0.01$

***$P < 0.001$.

immunotherapy have made remarkable advances and breakthroughs, but still with limited efficacy. Therefore, identifying reliable diagnostic biomarkers and effective therapeutic targets for GC continues to be a significant research hotspot. Recently, reports have shown that EDIL3 was abnormally expressed in various tumors and tightly related to tumor initiation and progression. In this study, we analyzed the expression and prognostic value of EDIL3 in GC. In addition, we utilized bioinformatics methods to explore the potential role of EDIL3 in GC.

The GEPIA database was used to investigate EDIL3 expression in GC. GEPIA analysis indicated that EDIL3 expression was elevated in GC tissues and positively associated with GC stages. Likewise, the data from our center also showed similar results. Survival analysis showed that high EDIL3 expression was an independent prognostic indicator for worse survival in GC. *In vitro* experiments indicated that EDIL3 promotes proliferation, invasion, and migration of GC cells, which corresponded with the results of a previous report on GC (*Zhang et al., 2020*). *Xia et al. (2015)* reported that EDIL3 can promote HCC invasion and migration through the induction of epithelial-mesenchymal transition (EMT). *Zou et al. (2009)* demonstrated that EDIL3 accelerates tumor growth by stimulating angiogenesis in colon cancer. The results of our research demonstrated that EDIL3 may play an essential role in the formation and development of GC, and is a potential marker for its diagnosis and prognosis.

Until recently, the mechanism of abnormal expression of EDIL3 in human cancer was not fully understood. Previous studies suggested that DNA methylation could regulate gene expression. Our study detected the abnormal DNA methylation of EDIL3 genes in various cancers. Methylation analysis revealed that CpG sites with higher methylation levels were mostly located in the Open_Sea regions of EDIL3. EDIL3 methylation at certain CpG sites was associated with worse prognosis in GC. Hence, the results suggested that the methylation levels of EDIL3 might be a potential prognostic biomarker for GC. Additional studies are required to explore the clinical role and significance of EDIL3 methylation in GC.

To further unravel the function of EDIL3 in cancer, GGI and PPI networks were constructed. The data showed that EDIL3 was mainly involved in cell adhesion-associated

functions and pathways. Moreover, co-expression analysis and functional enrichment analysis results indicated that most genes co-expressed with EDIL3 in GC were focused on ECM-receptor interaction and extracellular structure organization. As a secreted ECM protein, EDIL3 is involved in multiple processes in cancer occurrence and development. *Feng et al. (2014)* demonstrated that EDIL3 promotes anoikis resistance and anchorage independent growth advantage through the activation of integrin signal pathways. *Zhang et al. (2020)* found that EDIL3 was involved in cell migration, invasion, and EMT by regulating TGF-$\beta$1 signaling. *Gasca et al. (2020)* reported that EDIL3 may promote EMT through interactions with integrin $\alpha$V $\beta$3. Taken together, these results suggested that EDIL3 plays a crucial function in regulating tumor progression and metastasis *via* cell–matrix interactions.

In recent years, immunotherapy has shown great potential in GC. Previous studies reported that EDIL3 could modulate immunocyte adhesion through the binding of leukocyte-specific integrins (*Choi et al., 2008*). *Li et al. (2020b)* found that EDIL3 could promote regulatory T cell responses during inflammation resolution. These studies revealed that EDIL3 might impact cancer development and metastasis by affecting tumor immunity. Our results showed that EDIL3 was closely correlated with most immune cells and various immune cell markers in GC. Additionally, our study showed that EDIL3 expression was positively associated with several immune checkpoint markers and two MMR-related genes in GC. Moreover, EDIL3 may influence the prognosis of GC patients partially through immune cell infiltration. Consequently, we hypothesized that EDIL3 may affect the efficacy of immunotherapy and prognosis by modulating immune cell activity and immune-related gene expression in GC. The exact function of EDIL3 in the tumor-immune microenvironment needs further investigation.

Combined with above results, our data indicated that EDIL3 is a potential therapeutic target for GC. The results of drug sensitivity analysis showed that overexpression of EDIL3 in GC cells increased the sensitivity of 5-fluorouracil and paclitaxel. Similar to our findings, *Jia et al. (2021)* established a risk score based on EDIL3 expression, which was used to predict the sensitivity to chemotherapy drugs. The risk score was positively correlated with drug sensitivity to carboplatin and 5-fluorouracil in GC. Lapatinib, a dual EGFR and HER2neu tyrosine kinase inhibitor, is an effective agent used to treat HER2-positive breast cancer. However, the use of lapatinib in GC treatment is still controversial. S0413 research showed that single-agent lapatinib demonstrated limited activity in advanced GC (*Iqbal et al., 2011*), but clinical benefits were not shown in some other studies, such as the TyTAN study (*Satoh et al., 2014*) or TRIO-013/LOGiC study (*Hecht et al., 2016*). The drug sensitivity analysis revealed that EDIL3 expression was significantly correlated with lapatinib sensitivity in GC. High expression of EDIL3 may have a better response to lapatinib. These findings suggested that EDIL3 may be used as a therapeutic target or predictor of efficacy for GC patients.

## CONCLUSION

In conclusion, this study revealed that EDIL3 expression is elevated in GC and correlates with a worse prognosis. Additionally, this research also revealed that EDIL3 participates

in many malignant behaviors and affects immune cell infiltration in GC. These results suggested that EDIL3 plays an important role in the carcinogenesis and progression of GC. We speculated that EDIL3 might be a potential diagnostic and therapeutic target for GC. More studies are needed to elucidate the underlying molecular mechanism.

### Funding

This study was supported by grants from the National Natural Science Foundation of China (No. 81401952), the Science and Technology Research Projects of Tianjin Municipal Health Bureau (No. 2014KZ082), and the Tianjin Key Medical Discipline (Specialty) Construction Project (TJYXZDXK-009A). The funders had no role in study design, data collection and analysis, decision to publish, or preparation of the manuscript.

### Grant Disclosures

The following grant information was disclosed by the authors:
National Natural Science Foundation of China: 81401952.
Science and Technology Research Projects of Tianjin Municipal Health Bureau: 2014KZ082.
Tianjin Key Medical Discipline(Specialty) Construction Project: TJYXZDXK-009A.

### Competing Interests

The authors declare there are no competing interests.

### Author Contributions

- Bin Ke conceived and designed the experiments, performed the experiments, analyzed the data, prepared figures and/or tables, authored or reviewed drafts of the article, and approved the final draft.
- Zheng-Kai Liang performed the experiments, prepared figures and/or tables, and approved the final draft.
- Bin Li performed the experiments, prepared figures and/or tables, and approved the final draft.
- Xue-Jun Wang analyzed the data, prepared figures and/or tables, and approved the final draft.
- Ning Liu analyzed the data, authored or reviewed drafts of the article, and approved the final draft.
- Han Liang conceived and designed the experiments, authored or reviewed drafts of the article, and approved the final draft.
- Ru-Peng Zhang conceived and designed the experiments, authored or reviewed drafts of the article, and approved the final draft.

### Human Ethics

The following information was supplied relating to ethical approvals (*i.e.*, approving body and any reference numbers):

This study was ratified by the Ethics Committee of Tianjin Medical University Cancer Institute and Hospital.

## Data Availability

The raw measurements are available in the Supplemental Files.

## Supplemental Information

Supplemental information for this article can be found online at http://dx.doi.org/10.7717/peerj.15559#supplemental-information.

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
