# Peer review of "EDIL3 is a potential prognostic biomarker that correlates with immune infiltrates in gastric cancer"

_PeerJ, doi:10.7717/peerj.15559_

## Round 0.1 · original submission · Minor Revisions

Based on the reviewer's reports and my own evaluation article is worth publishing in the journal after minor revision.

Reviewer 1 ·

Basic reporting

The focus of this manuscript is to explore the role of EDIL3 in gastric cancer (GC). The authors utilized various bioinformatics resources and tools to evaluate different aspects of EDIL3 in GC, such as expression, methylation, prognostic value, etc. They also validated the in-silicon findings by in vitro experiments. Their study greatly advances the knowledge in the field. The manuscript is overall well-written. I only have a few suggestions for improvement.

Experimental design

1. The authors used qRT-PCR and western blot assay to validate the higher EDIL3 expression in GC than in normal tissues. 20 pairs of GC tumor tissues and adjacent non-neoplastic tissues were used, and the mRNA level of EDIL3 elevated in 13 of the tumor tissues. How many tumor tissues had an unregulated EDIL3 at the protein level? And did the 13 tumor tissues with elevated EDIL3 at mRNA level also have unregulated EDIL3 at the protein level?

2. The methods section lacks sufficient detail to replicate. Parameter used in various bioinformatics resources and tools should be provided.

Validity of the findings

1. For Fig. 1, why the two already published images (from HPA database) included here without proper references?

2. Figure7 and Figure 9 are not clear enough to allow the reader to receive any finding. Figure legends need to be more detailed. Scale bars are needed for Fig. 1G-H and Fig. 3D.

3. The manuscript needs to be proofread to remove grammar mistakes and ambiguous sentences. Some examples where the language could be improved include:
Lines 17-19: Epidermal Growth Factor-like repeats and Discoidin I-Like Domains 3 (EDIL3) is
a secretory protein that play important roles in embryonic development and various illnesses such as cancer.
Lines 55-57: Those studies indicated that the EDIL3 to be differentially expressed in different tumors and to play different functions in different types of tumor.
Line 58: Presently, the roles of EDIL3 in GC are remains largely elusive, and there are few studies on the function of EDIL3 in GC.

Reviewer 2 ·

Basic reporting

The article entitled ‘EDIL3 is a potential prognostic biomarker and 2 correlates with immune infiltrates in gastric cancer’ by Bin Ke et al., demonstrates the function of EDIL3 as an oncogene and suggests that EDIL3 may serve as a potential therapeutic target GC. Overall, the article is presented well, and the statement is validated with various experiments. The manuscript is suitable for publication with minor revisions.
1. Enlist discovered the function of EDIL3 in GC in lines 58-59.
2. If there are no limitations on words then please elaborate on the results section of various experiments, it appears that authors are a little shy to write the detail about the findings.
3. In the STRING relation network author has shown that expression of EDIL3 is directly associated with ITGAV, ITGB3, ITGB5, ZNF469, and PTK2. GEPIA. Although it is not recommended, if possible, the author may show the relationship between EDIL3 with any of the genes (ITGAV, ITGB3, ITGB5, ZNF469, and PTK2. GEPIA) with some wet lab techniques. Preferably at the mRNA expression level or cellular protein expression level.

Experimental design

.

Validity of the findings

.

Additional comments

.

---

## Round 0.2 · Minor Revisions

Based on one reviewer and my own evaluation figure 6 and figure 9 are not readable in their current form. Therefore, it is advisable to replace the figure with a high-quality image.

In addition, the Section Editor notes that the manuscript needs editing for English and/or clarity.

For example, "Epidermal Growth Factor-like repeats and Discoidin I-Like Domains 3 (EDIL3) is a secretory protein that plays important roles in embryonic development and various illnesses." Do you mean "are secretory proteins", or do you mean "Secretory proteins with Epidermal Growth Factor-like repeats and Discoidin I-Like Domains 3 (EDIL3) play important roles . . . "

The remainder of the manuscript needs to be edited for similar kinds of problems.

Reviewer 1 ·

Basic reporting

please see additional comments

Experimental design

please see additional comments

Validity of the findings

please see additional comments

Additional comments

The authors had clarified the relationship between RNA-seq and Western-blot results. The manuscript had also been revised to fix grammar mistakes. The authors had also deleted two published images (from HPA database).

The quality of the figures had been improved, however, it is still impossible to read figure 6 and figure 9 in their current quality.

Reviewer 2 ·

Basic reporting

.

Experimental design

.

Validity of the findings

.

Additional comments

The author's have addressed all the comments raised before. Now this manuscript should be accepted.

---

## Round 0.3 · Minor Revisions

Although, both reviewers have suggested that the manuscript has been revised substantially as per the comments/concerns raised. However, in some figures, their labeling should be used with a bigger font for visibility.

IN addition, some editing is still required:

Line 42, "Recently, extracellular matrix proteins have been implicated in cancer ..."

Line 29, start new paragraph "Recently, the role of EDIL3 in ..."

Line 255, "EDIL3 expression was increased in various" - you should say that the expression was increased in some tumors (not all show an increase)

In the next sentence, you say that EDIL3 was increased in GC, but was it significant (doesn't look so)

---

## Round 0.4 · accepted · Accept

I feel the authors have addressed the raised concerns adequately and the articles may be published soon.

In particular, the Section Editor suggests that it would be clearer to readers if you were to change the first line of the Abstract from:

> Epidermal growth factor-like repeats and discoidin I-like domains 3 (EDIL3) is a secretory protein that ...

to (for example)

> EDIL3, which contains epidermal growth factor-like repeats and discoidin I-like domains, is a secretory protein that plays...